# Learning Perceptual Inference by Contrasting

**Chi Zhang**[*,1,4], **Baoxiong Jia**[*,1], **Feng Gao**[3,4], **Yixin Zhu**[3,4], **Hongjing Lu**[2], **Song-Chun Zhu**[1,3,4]
[1] Department of Computer Science, University of California, Los Angeles
[2] Department of Psychology, University of California, Los Angeles
[3] Department of Statistics, University of California, Los Angeles
[4] International Center for AI and Robot Autonomy (CARA)
{chi.zhang,baoxiongjia,f.gao,yixin.zhu,hongjing,sczhu}@ucla.edu

## Abstract

"Thinking in pictures," [1] *i.e.*, spatial-temporal reasoning, effortless and instantaneous for humans, is believed to be a significant ability to perform logical induction and a crucial factor in the intellectual history of technology development. Modern Artificial Intelligence (AI), fueled by massive datasets, deeper models, and mighty computation, has come to a stage where (super-)human-level performances are observed in certain specific tasks. However, current AI's ability in "thinking in pictures" is still far lacking behind. In this work, we study how to improve machines' reasoning ability on one challenging task of this kind: Raven's Progressive Matrices (RPM). Specifically, we borrow the very idea of "contrast effects" from the field of psychology, cognition, and education to design and train a permutation-invariant model. Inspired by cognitive studies, we equip our model with a simple inference module that is jointly trained with the perception backbone. Combining all the elements, we propose the *Contrastive Perceptual Inference* network (CoPINet) and empirically demonstrate that CoPINet sets the new state-of-the-art for permutation-invariant models on two major datasets. We conclude that spatial-temporal reasoning depends on envisaging the possibilities consistent with the relations between objects and can be solved from pixel-level inputs.

## 1 Introduction

Among the broad spectrum of computer vision tasks are ones where dramatic progress has been witnessed, especially those involving visual information retrieval [2–5]. Significant improvement has also manifested itself in tasks associating visual and linguistic understanding [6–9]. However, it was only until recently that the research community started to re-investigate tasks relying heavily on the ability of "thinking in pictures" with modern AI approaches [1, 10, 11], particularly spatial-temporal inductive reasoning [12–14]; this line of work primarily focuses on Raven's Progressive Matrices (RPM) [15, 16]. It is believed that RPM is closely related to real intelligence [17], diagnostic of abstract and structural reasoning ability [18], and characterizes *fluid intelligence* [19–22]. In such a test, subjects are provided with two rows of figures following certain *unknown* rules and asked to pick the correct answer from the choices that would best complete the third row with a missing entry; see Figure 1(a) for an example. As shown in early works [12, 14], despite the fact that *visual elements* are relatively straightforward, there is still a notable performance gap between human and machine *visual reasoning* in this challenging task.

One missing ingredient that may result in this performance gap is a proper form of contrasting mechanism. Originated from perceptual learning [23, 24], it is well established in the field of psychology and education [25–29] that teaching new concepts by comparing with noisy examples is

---

[*] indicates equal contribution.

quite effective. Smith and Gentner [30] summarize that comparing cases facilitates transfer learning and problem-solving, as well as the ability to learn relational categories. Gentner [31] in his structure-mapping theory points out that learners generate a structure alignment between two representation when they compare two cases. A more recent study from Schwartz et al. [32] also shows that contrasting cases help foster an appreciation of a deep understanding of concepts.

We argue that such a *contrast effect* [33], found in both humans and animals [34–38], is essential to machines' reasoning ability as well. With access to how the data is generated, a recent attempt [13] finds that models demonstrate better generalizability if the choice of data and the manner in which it is presented to the model are made "contrastive." In this paper, we try to address a more direct and challenging question, *independent* of how the data is generated: how to incorporate an explicit contrasting mechanism during model *training* in order to improve machines' reasoning ability? Specifically, we come up with two levels of contrast in our model: a novel contrast module and a new contrast loss. At the model level, we design a permutation-invariant contrast module that summarizes the common features and distinguishes each candidate by projecting it onto its residual on the common feature space. At the objective level, we leverage ideas in contrastive estimation [39–41] and propose a variant of Noise-Contrastive Estimation (NCE) loss.

Another reason why RPM is challenging for existing machine reasoning systems could be attributed to the demanding nature of the *interplay* between perception and inference. Carpenter et al. [17] postulate that a proper understanding of one RPM instance requires not only an accurate encoding of individual elements and their visual attributes but also the correct induction of the hidden rules. In other words, to solve RPM, machine reasoning systems are expected to be equipped with *both* perception and inference subsystems; lacking either component would only result in a sub-optimal solution. While existing work primarily focuses on perception, we propose to bridge this gap with a simple inference module *jointly* trained with the perception backbone; specifically, the inference module reasons about which category the current problem instance falls into. Instead of training the inference module to predict the ground-truth category, we borrow the basis learning idea from [42] and jointly learn the inference subsystem with perception. This basis formulation could also be regarded as a hidden variable and trained using a log probability estimate.

Furthermore, we hope to make a critical improvement to the model design such that it is truly *permutation-invariant*. The invariance is mandatory, as an ideal RPM solver should not change the representation simply because the rows or columns of answer candidates are swapped or the order of the choices alters. This characteristic is an essential trait missed by all recent works [12, 14]. Specifically, Zhang et al. [12] stack all choices in the channel dimension and feed it into the network in one pass. Barrett et al. [14] add additional positional tagging to their Wild Relational Network (WReN). Both of them *explicitly* make models permutation-sensitive. We notice in our experiments that removing the positional tagging in WReN decreases the performance by $28\%$, indicating that the model bypasses the intrinsic complexity of RPM by remembering the positional association. Making the model permutation-invariant also shifts the problem from classification to ranking.

Combining contrasting, perceptual inference, and permutation invariance, we propose the *Contrastive Perceptual Inference* network (CoPINet). To verify its effectiveness, we conduct comprehensive experiments on two major datasets: the RAVEN dataset [12] and the PGM dataset [14]. Empirical studies show that our model achieves human-level performance on RAVEN and a new record on PGM, setting new state-of-the-art for permutation-invariant models on the two datasets. Further ablation on RAVEN and PGM reveals how each component contributes to performance improvement. We also investigate how the model performance varies under different sizes of datasets, as a step towards an ideal machine reasoning system capable of low-shot learning.

This paper makes four major contributions:
- We introduce two levels of contrast to improve machines' reasoning ability in RPM. At the model level, we design a contrast module that aggregates common features and projects each candidate to its residual. At the objective level, we use an NCE loss variant instead of the cross-entropy to encourage contrast effects.
- Inspired by Carpenter et al. [17], we incorporate an inference module to learn with the perception backbone jointly. Instead of using ground-truth, we regularize it with a fixed number of bases.
- We make our model permutation-invariant in terms of swapped rows or columns and shuffled answer candidates, shifting the previous view of RPM from classification to ranking.
- Combining ideas above, we propose CoPINet that sets new state-of-the-art on two major datasets.

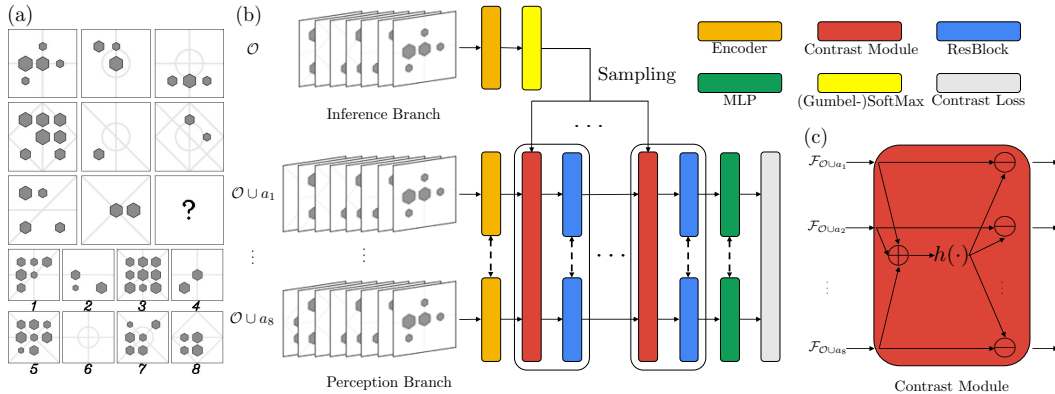

Figure 1: (a) An example of RPM. The hidden rule(s) in this problem can be denoted as $\{[\text{OR}, \text{line}, \text{type}]\}$, where an OR operation is applied to the type attribute of all lines, following the notations in Barrett et al. [14]. It is further noted that the OR operation is applied row-wise, and there is only one choice that satisfies the row-wise OR constraint. Hence the correct answer should be 5. (b) The proposed CoPINet architecture. Given a RPM problem, the inference branch samples a most likely rule for each attribute based only on the context $\mathcal{O}$ of the problem. Sampled rules are transformed and fed into each contrast module in the perception branch. Note that the combination of the contrast module and the residual block can be repeated. Dashed lines indicate that parameters are shared among the modules. (c) A sketch of the contrast module.

## 2  Related Work

**Contrastive Learning**   Teaching concepts by comparing cases, or contrasting, has proven effective in both human learning and machine learning. Gentner [31] postulates that human's learning-by-comparison process is a structural mapping and alignment process. A later article [43] firmly supports this conjecture and shows finding the individual difference is easier for humans when similar items are compared. Recently, Smith and Gentner [30] conclude that learning by comparing two contrastive cases facilitates the distinction between two complex interrelated relational concepts. Evidence in educational research further strengthens the importance of contrasting—quantitative structure of empirical phenomena is less demanding to learn when contrasting cases are used [32, 44, 45]. All the literature calls for a similar treatment of contrast in machine learning. While techniques from [46–48] are based on triplet loss using max margin to separate positive and negative samples, negative contrastive samples and negative sampling are proposed for language modeling [40] and word embedding [49, 50], respectively. Gutmann and Hyvärinen [39] discuss a general learning framework called Noise-Contrastive Estimation (NCE) for estimating parameters by taking noise samples into consideration, which Dai and Lin [41] follow to learn an effective image captioning model. A recent work [13] leverages contrastive learning in RPM; however, it focuses on data presentation while leaving the question of modeling and learning unanswered.

**Computational Models on RPM**   The cognitive science community is the first to investigate RPM with computational models. Assuming access to a perfect state representation, structure-mapping theory [31] and the high-level perception theory of analogy [51, 52] are designed with heuristics to solve the RPM problem at a symbolic level [17, 53–55]. Another stream of research approaches the problem by measuring the image similarity with hand-crafted state representations [56–60]. More recently, end-to-end data-driven methods with raw image input are proposed [12–14, 61]. Wang and Su [61] introduce an automatic RPM generation method. Barrett et al. [14] release the first large-scale RPM dataset and present a relational model [62] designed for it. Steenbrugge et al. [63] propose a pretrained $\beta$-VAE to improve the generalization performance of models on RPM. Zhang et al. [12] provide another dataset with structural annotations using stochastic image grammar [64–66]. Hill et al. [13] take a different approach and study how data presentation affects learning.

## 3  Learning Perceptual Inference by Contrasting

The task of RPM can be formally defined as: given a list of observed images $\mathcal{O} = \{o_i\}_{i=1}^{8}$, forming a $3 \times 3$ matrix with a final missing element, a solver aims to find an answer $a_\star$ from an *unordered* set

of choices $\mathcal{A} = \{a_i\}_{i=1}^8$ to best complete the matrix. Permutation invariance is a unique property for RPM problems: (1) According to [17], the same set of rules is applied either row-wise or column-wise. Therefore, swapping the first two rows or columns should not affect how one solves the problem. (2) In any multi-choice task, changing the order of answer candidates should not affect how one solves the problem either. These properties require us to use a permutation-invariant encoder and reformulate the problem from a typical classification problem into a ranking problem. Formally, in a probabilistic formulation, we seek to find a model such that

$$p(a_\star|\mathcal{O}) \geq p(a'|\mathcal{O}), \quad \forall a' \in \mathcal{A}, a' \neq a_\star, \tag{1}$$

where the probability is invariant when rows or columns in $\mathcal{O}$ are swapped. This formulation also calls for a model that produces a density estimation for each choice, regardless of its order in $\mathcal{A}$. To that end, we model the probability with a neural network equipped with a permutation-invariant encoder for each observation-candidate pair $f(\mathcal{O} \cup a)$. However, we argue such a purely perceptive system is far from sufficient without contrasting and perceptual inference.

## 3.1 Contrasting

To provide the reasoning system with a mechanism of contrasting, we propose to explicitly build two levels of contrast: model-level contrast and objective-level contrast.

### 3.1.1 Model-level Contrast

As the central notion of contrast is comparing cases [30, 32, 44, 45], we propose an explicit model-level contrasting mechanism in the following form,

$$\text{Contrast}(\mathcal{F}_{\mathcal{O} \cup a}) = \mathcal{F}_{\mathcal{O} \cup a} - h\left(\sum_{a' \in \mathcal{A}} \mathcal{F}_{\mathcal{O} \cup a'}\right), \tag{2}$$

where $\mathcal{F}$ denotes features of a specific combination and $h(\cdot)$ summarizes the common features in all candidate answers. In our experiments, $h(\cdot)$ is a composition of BatchNorm [67] and Conv.

Intuitively, this explicit contrasting computation enables a reasoning system to tell distinguishing features for each candidate in terms of fitting and following the rules hidden among all panels in the incomplete matrix. The philosophy behind this design is to constrain the functional form of the model to capture both the commonality and the difference in each instance. It is expected that the very inductive bias on comparing similarity and distinctness is baked into the entire reasoning system such that learning in the challenging task becomes easier.

In a generalized setting, each $\mathcal{O} \cup a$ could be abstracted out as an object. Then the design becomes a general contrast module, where each object is distinguished by comparing with the common features extracted from an object set.

We further note that the contrasting computation can be encapsulated into a single neural module and repeated: the addition and transformation are shared and the subtraction is performed on each individual element. See Figure 1(c) for a sketch of the contrast module. After such operations, permutation invariance of a model will not be broken.

### 3.1.2 Objective-level Contrast

To further enforce the contrast effects, we propose to use an NCE variant rather than the cross-entropy loss commonly used in previous works [12, 14]. While there are several ways to model the probability in Equation 1, we use a Gibbs distribution in this work:

$$p(a|\mathcal{O}) = \frac{1}{Z}\exp(f(\mathcal{O} \cup a)), \tag{3}$$

where $Z$ is the partition function, and our model $f(\cdot)$ corresponds to the negative potential function. Note that such a distribution has been widely adopted in image generation models [68–70].

In this case, we can take the log of both sides in Equation 1 and rearrange terms:

$$\log p(a_\star|\mathcal{O}) - \log p(a'|\mathcal{O}) = f(\mathcal{O} \cup a_\star) - f(\mathcal{O} \cup a') \geq 0, \quad \forall a' \in \mathcal{A}, a' \neq a_\star. \tag{4}$$

This formulation could potentially lead to a max margin loss. However, we notice in our preliminary experiments that max margin is not sufficient; we realize it is inferior to make the negative potential of the wrong choices only *slightly lower*. Instead, we would like to further push the difference to *infinity*. To do that, we leverage the *sigmoid* function $\sigma(\cdot)$ and train the model, such that:

$$f(\mathcal{O} \cup a_\star) - f(\mathcal{O} \cup a') \to \infty \iff \sigma(f(\mathcal{O} \cup a_\star) - f(\mathcal{O} \cup a')) \to 1, \forall a' \in \mathcal{A}, a' \neq a_\star. \quad (5)$$

However, we notice that the relative difference of negative potential is still problematic. We hypothesize this deficiency is due to the lack of a baseline—without such a regularization, the negative potential of wrong choices could still be very high, resulting in difficulties in learning the negative potential of the correct answer. To this end, we modify Equation 5 into its sufficient conditions:

$$f(\mathcal{O} \cup a_\star) - b(\mathcal{O} \cup a_\star) \to \infty \iff \sigma(f(\mathcal{O} \cup a_\star) - b(\mathcal{O} \cup a_\star)) \to 1 \quad (6)$$

$$f(\mathcal{O} \cup a') - b(\mathcal{O} \cup a') \to -\infty \iff \sigma(f(\mathcal{O} \cup a') - b(\mathcal{O} \cup a')) \to 0, \quad (7)$$

where $b(\cdot)$ is a fixed baseline function and $a' \in \mathcal{A}, a' \neq a_\star$. For implementation, $b(\cdot)$ could be either a randomly initialized network or a constant. Since the two settings do not produce significantly different results in our preliminary experiments, we set $b(\cdot)$ to be a constant to reduce computation.

We then optimize the network to maximize the following objective as done in [39]:

$$\ell = \log(\sigma(f(\mathcal{O} \cup a_\star) - b(\mathcal{O} \cup a_\star))) + \sum_{a' \in \mathcal{A}, a' \neq a_\star} \log(1 - \sigma(f(\mathcal{O} \cup a') - b(\mathcal{O} \cup a'))). \quad (8)$$

**Connection to NCE** If we treat the baseline as the negative potential of a fixed noise model of the same Gibbs form and ignore the difference between the partition functions, Equation 6 and Equation 7 become the $G$ function used in NCE [39]. But unlike NCE, we do not need to multiply the size ratio in the sigmoid function [41].

### 3.2 Perceptual Inference

As indicated in Carpenter et al. [17], a mere perceptive model for RPM is arguably not enough. Therefore, we propose to incorporate a simple inference subsystem into the model: the inference branch should be responsible for inferring the hidden rules in the problem. Specifically, we assume there are at most $N$ attributes in each problem, each of which is subject to the governance of one of $M$ rules. Then hidden rules $\mathcal{T}$ in one problem instance can be decomposed into

$$p(\mathcal{T}|\mathcal{O}) = \prod_{i=1}^{N} p(t_i|\mathcal{O}), \quad (9)$$

where $t_i = 1 \ldots M$ denotes the rule type on attribute $n_i$. For the actual form of the probability of rules on each attribute, we propose to model it using a multinomial distribution. This assumption is consistent with the way datasets are usually generated [12, 14, 61]: one rule is independently picked from the rule set for each attribute. In this way, each rule could also be regarded as a basis in a rule dictionary and jointly learned, as done in active basis [42] or word embedding [49, 71].

If we treat rules as hidden variables, the log probability in Equation 4 can be decomposed into

$$\log p(a|\mathcal{O}) = \log \sum_{\mathcal{T}} p(a|\mathcal{T}, \mathcal{O})p(\mathcal{T}|\mathcal{O}) = \log \mathbb{E}_{\mathcal{T} \sim p(\mathcal{T}|\mathcal{O})}[p(a|\mathcal{T}, \mathcal{O})]. \quad (10)$$

Note that writing the summation in the form of expectation affords sampling algorithms, which can be done on each individual attribute due to the independence assumption.

In addition, if we model $p(\mathcal{T}|\mathcal{O})$ as an inference branch $g(\cdot)$ and sample only once from it, the model can be modified into $f(\mathcal{O} \cup a, \hat{\mathcal{T}})$ with $\hat{\mathcal{T}}$ sampled from $g(\mathcal{O})$. Following the same derivation above, we now optimize the new objective:

$$\ell = \log(\sigma(f(\mathcal{O} \cup a_\star, \hat{\mathcal{T}}) - b(\mathcal{O} \cup a_\star))) + \sum_{a' \in \mathcal{A}, a' \neq a_\star} \log(1 - \sigma(f(\mathcal{O} \cup a', \hat{\mathcal{T}}) - b(\mathcal{O} \cup a'))). \quad (11)$$

To sample from a multinomial, we could either use hard sampling like Gumbel-SoftMax [72, 73] or a soft one by taking expectation. We do not observe significant difference between the two settings.

The expectation in Equation 10 is proposed primarily to make the computation of the exact log probability controllable and tractable: while the full summation requires $O(M^N)$ passes of the model, a Monte Carlo approximation of it could be calculated in $O(1)$ time. We also note that if $p(\mathcal{T}|\mathcal{O})$ is highly peaked (*e.g.*, ground truth), the Monte Carlo estimate could be accurate as well. Despite the fact that we only sample once from an inference branch to reduce computation, we find in practice the Monte Carlo estimate works quite well.

### 3.3 Architecture

Combining contrasting, perceptual inference, and permutation invariance, we propose a new network architecture to solve the challenging RPM problem, named *Contrastive Perceptual Inference* network (CoPINet). The perception branch is composed of a common feature encoder and shared interweaving contrast modules and residual blocks [3]. The encoder first extracts image features independently for each panel and sum ones in the corresponding rows and columns before the final transformation into a latent space. The inference branch consists of the same encoder and a (Gumbel-)SoftMax output layer. The sampled results will be transformed and concatenated channel-wise into the summation in Equation 2. In our implementation, we prepend each residual block with a contrast module; such a combination can be repeated while keeping the network permutation-invariant. The network finally uses an MLP to produce a negative potential for each observation and candidate pair and is trained using Equation 11; see Figure 1(b) for a graphical illustration of the entire CoPINet architecture.

## 4 Experiments

### 4.1 Experimental Setup

We verify the effectiveness of our models on two major RPM datasets: RAVEN [12] and PGM [14]. Across all experiments, we train models on the training set, tune hyper-parameters on the validation set, and report the final results on the test set. All of the models are implemented in PyTorch [74] and optimized using ADAM [75]. While a good performance of WReN [14] and ResNet+DRT [12] relies on external supervision, such as rule specifications and structural annotations, the proposed model achieves better performance with only $\mathcal{O}$, $\mathcal{A}$, and $a_\star$. Models are trained on servers with four Nvidia RTX Titans. For the WReN model, we use a public implementation that reproduces results in [14][1]. We implement our models in PyTorch [74] and optimize using ADAM [75]. During training, we perform early-stop based on validation loss. We use the same network architecture and hyper-parameters in both RAVEN and PGM experiments.

### 4.2 Results on RAVEN

There are $70,000$ problems in the RAVEN dataset [12], equally distributed in 7 figure configurations. In each configuration, the dataset is randomly split into 6 folds for training, 2 folds for validation, and 2 folds for testing. We compare our model with several simple baselines (LSTM [76], CNN [77], and vanilla ResNet [3]) and two strong baselines (WReN [14] and ResNet+DRT [12]). Model performance is measured by accuracy.

**General Performance on RAVEN**    In this experiment, we train the models on all $42,000$ training samples and measure how they perform on the test set. The first part of Table 1 shows the testing accuracy of all models. We also retrieve the performance of humans and a solver with perfect information from [12] for comparison. As shown in the table, the proposed model CoPINet achieves the best performance among all the models we test. For the relational model WReN proposed in [14], we run the tests on a permutation-invariant version, *i.e.*, one without positional tagging (NoTag), and tune the model also to minimize an auxiliary loss (Aux) [14]. While the auxiliary loss could boost the performance of WReN as we will show later in the ablation study, we do not observe similar effects on CoPINet. As indicated in the detailed comparisons in Table 1, WReN is biased towards images of grid configurations and does poorly on ones demanding compositional reasoning, *i.e.*, ones with independent components. We further note that compared to previously proposed models (WReN [14] and ResNet+DRT [12]), CoPINet does not require additional information such as structural annotations and meta targets and still shows human-level performance in this task. When

comparing the performance of CoPINet and human on specific figure configurations, we notice that CoPINet is inferior in learning samples of grid-like compositionality but efficient in distinguishing images consisting of multiple components, implying the efficiency of the contrasting mechanism.

**Ablation Study**   One problem of particular interest in building CoPINet is how each component contributes to performance improvement. To answer this question, we measure model accuracy by gradually removing each construct in CoPINet, *i.e.*, the perceptual inference branch, the contrast loss, and the contrast module. In the second part of Table 1, we show the results of ablation on CoPINet. Both the full model (CoPINet) and the one without the perceptual inference branch (CoPINet-Contrast-CL) could achieve human-level performance, with the latter slightly inferior to the former. If we further replace the contrast loss with the cross-entropy loss (CoPINet-Contrast-XE), we observe a noticeable performance decrease of around $4\%$, verifying the effectiveness of the contrast loss. A catastrophic performance downgrade of $66\%$ is observed if we remove the contrast module, leaving only the network backbone (CoPINet-Backbone-XE). This drastic performance gap shows that the functional constraint on modeling an explicit contrasting mechanism is arguably a crucial factor in machines' reasoning ability as well as in humans'. The ablation study shows that all the three proposed constructs, especially the contrast module, are critical to the performance of CoPINet. We also study how the requirement of permutation invariance and auxiliary training affect the previously proposed WReN. As shown in Table 1, sacrificing the permutation invariance (Tag) provides the model a huge upgrade during auxiliary training (Aux), compared to the one without tagging (NoTag) and auxiliary loss (NoAux). This effect becomes even more significant on the PGM dataset, as we will show in Section 4.3.

**Dataset Size and Performance**   Even though CoPINet surpasses human performance on RAVEN, this competition is inherently unfair, as the human subjects in this study never experience such an intensive training session as our model does. To make the comparison fairer and also as a step towards a model capable of human learning efficiency, we further measure how the model performance changes as the training set size shrinks. To this end, we train our CoPINet on subsets of the full RAVEN training set and test it on the full test set. As shown on Table 2 and Figure 2, the model performance varies roughly log-linearly with the training set size. One surprising observation is: with only half of the amount of the data, we could already achieve human-level performance. On a training set $16\times$ smaller, CoPINet outperforms all previous models. And on a subset $64\times$ smaller, CoPINet already outshines WReN.

## 4.3   Results on PGM

We use the neutral regime of the PGM dataset for model evaluation due to its diversity and richness in relationships, objects, and attributes. This split of the dataset has in total $1.42$ million samples, with $1.2$ million for training, $2,000$ for validation, and $200,000$ for testing. We train the models on the training set, tune the hyperparameters on the validation set, and evaluate the performance on the test

Table 1: Testing accuracy of models on RAVEN. Acc denotes the mean accuracy of each model. Same as in [12], L-R denotes the Left-Right configuration, U-D Up-Down, O-IC Out-InCenter, and O-IG Out-InGrid.

| Method | Acc | Center | 2x2Grid | 3x3Grid | L-R | U-D | O-IC | O-IG |
|---|---|---|---|---|---|---|---|---|
| LSTM | 13.07% | 13.19% | 14.13% | 13.69% | 12.84% | 12.35% | 12.15% | 12.99% |
| WReN-NoTag-Aux | 17.62% | 17.66% | 29.02% | 34.67% | 7.69% | 7.89% | 12.30% | 13.94% |
| CNN | 36.97% | 33.58% | 30.30% | 33.53% | 39.43% | 41.26% | 43.20% | 37.54% |
| ResNet | 53.43% | 52.82% | 41.86% | 44.29% | 58.77% | 60.16% | 63.19% | 53.12% |
| ResNet+DRT | 59.56% | 58.08% | 46.53% | 50.40% | 65.82% | 67.11% | 69.09% | 60.11% |
| CoPINet | **91.42**% | **95.05**% | **77.45**% | **78.85**% | **99.10**% | **99.65**% | **98.50**% | **91.35**% |
| WReN-NoTag-NoAux | 15.07% | 12.30% | 28.62% | 29.22% | 7.20% | 6.55% | 8.33% | 13.10% |
| WReN-Tag-NoAux | 17.94% | 15.38% | 29.81% | 32.94% | 11.06% | 10.96% | 11.06% | 14.54% |
| WReN-Tag-Aux | 33.97% | 58.38% | 38.89% | 37.70% | 21.58% | 19.74% | 38.84% | 22.57% |
| CoPINet-Backbone-XE | 20.75% | 24.00% | 23.25% | 23.05% | 15.00% | 13.90% | 21.25% | 24.80% |
| CoPINet-Contrast-XE | 86.16% | 87.25% | 71.05% | 74.45% | 97.25% | 97.05% | 93.20% | 82.90% |
| CoPINet-Contrast-CL | 90.04% | 94.30% | 74.00% | 76.85% | 99.05% | 99.35% | 98.00% | 88.70% |
| Human | 84.41% | 95.45% | 81.82% | 79.55% | 86.36% | 81.81% | 86.36% | 81.81% |
| Solver | 100% | 100% | 100% | 100% | 100% | 100% | 100% | 100% |

Figure 2: CoPINet on RAVEN and PGM as the training set size shrinks.

Table 2: Model performance under different training set sizes on RAVEN dataset. The full training set has 42,000 samples.

Table 3: Model performance under different training set sizes on PGM dataset. The full training set has 1.2 million samples.

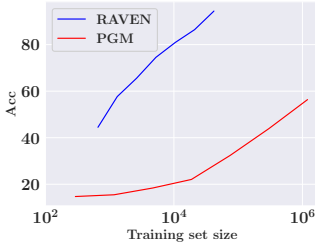

| Training set size | Acc |
|---|---|
| 658 | 44.48% |
| 1,316 | 57.69% |
| 2,625 | 65.55% |
| 5,250 | 74.53% |
| 10,500 | 80.92% |
| 21,000 | 86.43% |

| Training set size | Acc |
|---|---|
| 293 | 14.73% |
| 1,172 | 15.48% |
| 4,688 | 18.39% |
| 18,750 | 22.07% |
| 75,000 | 32.39% |
| 300,000 | 43.89% |

Table 4: Testing accuracy of models on PGM. Acc denotes the mean accuracy of each model.

| Method | CNN | LSTM | ResNet | Wild-ResNet | WReN-NoTag-Aux | CoPINet |
|---|---|---|---|---|---|---|
| Acc | 33.00% | 35.80% | 42.00% | 48.00% | 49.10% | **56.37%** |

set. We compare our models with baselines set up in [14], *i.e.*, LSTM, CNN, ResNet, Wild-ResNet, and WReN. As ResNet+DRT proposed in [12] requires structural annotations not available in PGM, we are unable to measure its performance. Again, all performance is measured by accuracy. Due to the lack of further stratification on this training regime, we only report the final mean accuracy.

**General Performance on PGM** In this experiment, we train the models on all 1.2 million training samples and report performance on the entire test set. As shown in Table 4, CoPINet achieves the best performance among all permutation-invariant models, setting a new state-of-the-art on this dataset. Similar to the setting in RAVEN, we make the previously proposed WReN permutation-invariant by removing the positional tagging (NoTag) and train it with both cross-entropy loss and auxiliary loss (Aux) [14]. The auxiliary loss could boost the performance of WReN. However, in coherence with the study on RAVEN and a previous work [12], we notice that the auxiliary loss does not help our CoPINet. It is worth noting that while WReN demands additional training supervision from meta targets to reach the performance, CoPINet only requires basic annotations of ground truth indices $a_\star$ and achieves better results.

**Ablation Study** We perform ablation studies on both WReN and CoPINet to see how the requirement of permutation invariance affects WReN and how each module in CoPINet contributes to its superior performance. The notations are the same as those used in the ablation study for RAVEN. As shown in the first part of Table 5, adding a proper auxiliary loss does provide WReN a 10% performance boost. However, additional supervision is required. Making the model permutation-sensitive gives the model a significant benefit by up to a 28% accuracy increase; however, it also indicates that WReN learns to shortcut the solutions by coding the positional association, instead of truly understanding the differences among distinctive choices and their potential effects on the compatibility of the entire matrix. The second part of Table 5 demonstrates how each construct contributes to the performance improvement of CoPINet on PGM. Despite the smaller enhancement of the contrast loss compared to that in RAVEN, the upgrade from the contrast module for PGM is still significant, and the perceptual inference branch keeps raising the final performance. In accordance with the ablation study on the RAVEN dataset, we show that all the proposed components contribute to the final performance increase.

Table 5: Ablation study on PGM.

| Method | WReN-NoTag-NoAux | WReN-NoTag-Aux | WReN-Tag-NoAux | WReN-Tag-Aux |
|---|---|---|---|---|
| Acc | 39.25% | 49.10% | 62.45% | 77.94% |

| Method | CoPINet-Backbone-XE | CoPINet-Contrast-XE | CoPINet-Contrast-CL | CoPINet |
|---|---|---|---|---|
| Acc | 42.10% | 51.04% | 54.19% | 56.37% |

**Dataset Size and Performance**    Motivated by the idea of fairer comparison and low-shot reasoning, we also measure how the performance of the proposed CoPINet changes as the training set size of PGM varies. Specifically, we train CoPINet on subsets of the PGM training set and test it on the entire test set. As shown in Table 3 and Figure 2, CoPINet performance on PGM varies roughly log-exponentially with respect to the training set size. We further note that when trained on a $16\times$ smaller dataset, CoPINet already achieves results similar to CNN and LSTM.

## 5    Conclusion and Discussion

In this work, we aim to improve machines' reasoning ability in "thinking in pictures" by jointly learning perception and inference via contrasting. Specifically, we introduce the contrast module, the contrast loss, and the joint system of perceptual inference. We also require our model to be permutation-invariant. In a typical and challenging task of this kind, Raven's Progressive Matrices (RPM), we demonstrate that our proposed model—*Contrastive Perceptual Inference* network (CoPINet)—achieves the new state-of-the-art for permutation-invariant models on two major RPM datasets. Further ablation studies show that all the three proposed components are effective towards improving the final results, especially the contrast module. It also shows that the permutation invariance forces the model to understand the effects of different choices on the compatibility of an entire RPM matrix, rather than remembering the positional association and shortcutting the solutions.

While it is encouraging to see the performance improvement of the proposed ideas on two big datasets, it is the last part of the experiments, *i.e.*, dataset size and performance, that really intrigues us. With infinitely large datasets that cover the entirety of an arbitrarily complex problem domain, it is arguably possible that a simple over-parameterized model could solve it. However, in reality, there is barely any chance that one would observe all the domain, yet humans still learn quite efficiently how the hidden rules work. We believe this is the core where the real intelligence lies: learning from only a few samples and generalizing to the extreme. Even though CoPINet already demonstrates better learning efficiency, it would be ideal to have models capable of few-shot learning in the task of RPM. Without massive datasets, it would be a real challenge, and we hope the paper could call for future research into it.

Performance, however, is definitely not the end goal in the line of research on relational and analogical visual reasoning: other dimensions for measurements include generalization, generability, and transferability. Is it possible for a model to be trained on a single configuration and generalize to other settings? Can we generate the final answer based on the given context panels, in a similar way to the top-down and bottom-up method jointly applied by humans for reasoning? Can we transfer the relational and geometric knowledge required in the reasoning task from other tasks? Questions like these are far from being answered. While Zhang et al. [12] show in the experiments that neural models do possess a certain degree of generalizability, the testing accuracy is far from satisfactory. In the meantime, there are a plethora of discriminative approaches towards solving reasoning problems in question answering, but generative methods and combined methods are lacking. The relational and analogical reasoning was initially introduced as a way to measure a human's intelligence, without training humans on the task. However, current settings uniformly reformulate it as a learning problem rather than a transfer problem, contradictory to why the task was started. Up to now, there has been barely any work that measures how knowledge on another task could be transferred to this one. We believe that significant advances in these dimensions would possibly enable Artificial Intelligence (AI) models to go beyond data fitting and acquire symbolized knowledge.

While modern computer vision techniques to solve Raven's Progressive Matrices (RPM) are based on neural networks, a promising ingredient is nowhere to be found: Gestalt psychology. Traces of the perceptual grouping and figure-ground organization are gradually faded out in the most recent wave of deep learning. However, the principles of grouping, both classical (*e.g.*, proximity, closure, and similarity) and new (*e.g.*, synchrony, element, and uniform connectedness) play an essential role in RPM, as humans arguably solve these problems by first figuring out groups and then applying the rules. We anticipate that modern deep learning methods integrated with the tradition of conceptual and theoretical foundations of the Gestalt approach would further improve models on abstract reasoning tasks like RPM.

**Acknowledgments:**  This work reported herein is supported by MURI ONR N00014-16-1-2007, DARPA XAI N66001-17-2-4029, ONR N00014-19-1-2153, NSF BSC-1827374, and an NVIDIA GPU donation grant.

## Footnotes

[1] `https://github.com/Fen9/WReN`

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
