[Reviews · NeurIPS 2019]

Reviewer 1



Originality: I find the originality to be good. There is a nice translation of contrastive learning from cognitive science and I believe the authors implemented it well in a machine learning setting. The contrast module seems novel to the best of my knowledge. Quality: I find the submission to be technically sound. The ablation study really shows the impact of the authors contrast module. Clarity: I found the article to be clear with respect to its own model, but lacking details when it came to the WReN permutation sensitivity. Significance: I find the significance of the authors empirical results to be high. RPM is a task that is well established at top-tier conferences and obtaining such strong SOTA results is very impressive. - Overall the authors make a valid point that a machine learning reasoning model should exhibit some sort of invariance with respect to the order of the multiple choice answers. In Lines 64-68 the authors claim that they removed positional tagging from one model WReN [14] and they claim it decreased performance by 28%. - For WReN: How exactly was this removal of positional tagging performed? Some more details would be very helpful here (even if it’s relegated to the Supplementary Materials). For example: exactly what are the WReN representations, how did you remove them, did you retrain, if so what was your training scheme, what type of hyperparameter search did you perform, etc. These would have been nice to have for completeness (say in the Supplementary Materials). - How exactly is WReN permutation sensitive? Please correct me if I am wrong but it is my understanding that WReN independently processes 8 different combinations O_1 = {O union a_1}, …, O_8 = {U union a_8}, where O is the 8 given panel sequence, and a_i are the multiple choice answers. Each O_i set contains 9 panels, and each panel in O_i is given a one-hot positional embedding. But if this is the case, then the a_i is always given the positional embedding of [0,…,0,1] e.g. all 0 except a 1 in the 9th coordinate. I quote from [14]: “The input vector representations were produced by processing each panel independently through a small CNN and tagging it with a panel label, similar to the LSTM processing described above…” and for the LSTM processing guideline: “Since LSTMs are designed to process inputs sequentially, we first passed each panel (context panels and multiple choice panels) sequentially and independently through a small 4-layer CNN, and tagged the CNN’s output with a one-hot label indicating the panel’s position (the top left PGM panel is tagged with label 1, the top-middle PGM panel is tagged with label 2, etc.) and passed the resulting sequence of labelled embeddings to the LSTM.” If my understanding is correct, then I do not see how permuting the answer choices affects the WReN model in the spirit of Lines 118-120. Can the authors clarify here? - For [12] I believe models that stack all the answer choices along a channel axis should be evaluated via their permutation equivariance, not permutation invariance. - For completeness the authors are missing a reference to “Improving Generalization for Abstract Reasoning Tasks Using Disentangled Feature Representations” which also benchmarks Ravens Progressive Matrices task. - The contrast module in Figure 1c is well diagrammed and intuitive to understand. One of the benefits that immediately comes to mind for the contrast module vs. the RN model is that the contrast module seems to scale linearly in the number of answer choices vs. the RN which produces a quadratic set. - The benchmarks on RPM are impressive and on PGM the authors are able to exhibit a very healthy gap in performance over “permutation invariant”-ized baseline models. The ablation studies really show how significant the contrast module is. - Overall I view this submission in the following manner: there is clear architectural novelty, the motivation for permutation invariance is obvious, and the results are quite strong. I’m convinced that the authors model clearly beats out the WReN model, but I believe the authors may have misrepresented the WReN model’s permutation sensitivity (see my above point about tagging of the panels). Specifically, I find the author’s use of removing the positional tagging to be misrepresentative of the permutation sensitivity of the WReN model. I would be happy if the authors corrected any misunderstanding on my part because I do believe the empirical results are strong and the hypothesis is intuitive and well-motivated. Grammar and expository-oriented comments: - Line 8: I don’t think this is the proper use of “i.e.” which is typically used like “in other words”. Instead it could be: “In this work, we study how to improve machines’ reasoning ability on one challenging task of this kind: the Raven’s Progressive Matrices (RPM).” - Line 306-307 is a bit too strong.

Reviewer 2



The paper feels a bit unprincipled. It combines three ideas. But how general these ideas are beyond RPMs is unclear to me. For example, perceptual inference usually means something rather different outside of RPMs. I read the paper a few times, mostly coming from the cognitive and DL field and it just did not feel overly enlightening. I am a bit unclear to which level RPM meaningfully can be described as reasoning with pictures. Above all, the solution produced by the authors does not feel like it has much of a reasoning flavor.

Reviewer 3



This paper was a pleasure to read and I very much recommend acceptance. Orginality: The work is original. It contributes a novel model which takes a ranking rather than classification perspective on the problem; a nice derivation of a NCE-loss variant; and permutation-invariance. It compares to the related work e.g WReN. Quality: The technical exposition is excellent and clear. I didn't see any issues. Clarity: Very well-written and enjoyable to read. I particularly appreciated that they author's presented their model (e.g. the object-level contrastive module) somewhat narratively, describing what they tried and why it didn't work and had to be adjusted in various ways. Significance: Pushes the envelope on this kind of analogical reasoning so a very strong contribution.

[Author Response · NeurIPS 2019]

We'd like to express our gratitude towards all the reviewers who have devoted their time to evaluating our paper and providing constructive feedback. Specifically, we'd like to thank Reviewer #1 (R1) and Reviewer #3 (R3) for acknowledging our novelty. We feel honored to receive such a high rating and compliments from R3 and we appreciate it that R1 points out the linear complexity of our contrastive module compared to that of the relational module which is quadratic. We will also take advice from Reviewer #2 (R2) and improve our paper in revision. The rest of the rebuttal will focus on addressing concerns from both R1 and R2.

**R1: Permutation in RPM, permutation-equivariance, and permutation-invariance**  We'll further clarify these terms in revision. Permutation is a unique property for RPM problems: (1) According to [17], in an RPM instance, the same set of rules is applied either row-wise or column-wise. Therefore, in a row-wise / column-wise instance, swapping the first two rows / columns should not affect how one solves the problem. (2) In any multi-choice task, changing the order of answer candidates should not affect how one solves the problem either. Permutation-equivariance refers to the case where a permutation applied to the input results in the same permutation on the corresponding output of a function. Permutation-invariance refers to the case where a permutation applied to the input does not change the output of a function. In this work, we hope that no matter how rows / columns are swapped and no matter how candidate answers are permuted, our computation process will always pick **the same** correct image, **rather than its index**. Hence, we consistently use "permutation-invariance" and always measure whether the correct image is picked, **not its index**.

**R1: WReN's representation and its permutation-sensitivity**  The representation of images and their positions in WReN is: each image's features are first extracted by a small CNN independently and then concatenated with a one-hot positional tagging of length 9 before further processing. WReN is permutation-invariant in the sense of (2) but not in (1). Since the context panels are tagged with their positions, when rows / columns are swapped, the final representation will be changed. For example, in an instance where rules are applied row-wise, swapping the first two rows will change tags of images originally in the first row from $[1, 2, 3]$ to $[4, 5, 6]$. WReN's tagging strategy will couple the same image representation with a different position embedding, generating different scores for the choices and making the model permutation-sensitive. In CoPINet, we use a shared Conv layer to independently extract image features and sum them together to avoid the problem. We are sorry that Line 118-120 is oversimplified. And we will use the additional page to discuss permutation in RPM and why WReN is permutation-sensitive in this sense. We will also make Figure 1 clearer to show how permutation-sensitivity is avoided in CoPINet.

**R1: Training of WReN-NoTag**  When evaluating the performance of WReN-NoTag, we did not concatenate positional taggings to the image features and trained the entire model from scratch. In this way, positions were no longer related to image features, and permutation problems from both (1) and (2) were avoided, hence the permutation-invariance; note that the relational module itself is permutation-invariant in the sense of both (1) and (2). We performed grid search to find the best hyper-parameters: batch size, learning rate, and weight of the auxiliary loss. We promise to release the code upon paper acceptance to benefit the community for future research in analogical reasoning.

**R1: Missing reference**  Thanks for pointing out. We will add in Line 112, between the two sentences, "Steenbrugge *et al*. [75] propose a novel training strategy to improve the generalization performance of models on RPM, where a $\beta$-VAE is pretrained to unsupervisedly learn a relational latent space and fine-tuned together with the model".

**R1: Grammar and wording**  We will change Line 8 into what is suggested by R1, check the entire manuscript, and correct all misuses of "*i.e.*". We will also change the verb in Line 306-307 from "lift" to "improve".

**R2: The paper is unprincipled as it just combines a few ideas. It doesn't feel like it has much of a reasoning flavor**  The insight of this paper is to incorporate contrasting together with inference. This insight for analogical reasoning has been firmly established in the literature [21-30], but rarely adopted in machine learning. In this paper, the contrast idea is implemented as the contrast module and the contrast loss, and inference refers to "a simple inference module jointly trained with the perception backbone" (Line 54-56), hence the "perceptual inference" in our paper title.

**R2: How can these ideas generalize beyond RPM**  Two important messages that can generalize for other analogical reasoning tasks are: (1) Contrasting, as demonstrated in the previous psychology literature, is indeed crucial to improve the performance of analogical reasoning for machine intelligence. The contrast module proposed in this paper could be easily plugged-in to any models for reasoning, ranking, or other discriminative learning, while the loss could be tested similarly on these tasks. (2) Improvement from the inference module suggests that we could use a sampling technique, let the model learn end-to-end by itself, and enjoy the performance boost.

**R2: How can RPM be described as reasoning with pictures**  Existing works describe the image-based RPM either as an abstract reasoning task [14] or as a relational and analogical reasoning task [12]. As discussed in [17], one needs to reason about what the hidden rule is from correct encodings of a limited number of image examples to solve the task. Our solution tries to improve current machine learning models on this challenging reasoning task.

[75] Xander Steenbrugge, Sam Leroux, Tim Verbelen, and Bart Dhoedt. Improving generalization for abstract reasoning tasks using disentangled feature representations.arXiv preprint arXiv:1811.04784, 2018.


[Meta-Review · NeurIPS 2019]

There is general agreement upon accepting this paper. The work helps build some bridges between communities.